Subject Areas:
evolution/palaeontology/palaeontology

Keywords:
endocast, brain, sarcopterygii, lungfish, coelacanth, tetrapod

Author for correspondence:
T. J. Challands
e-mail: tom.challands@ed.ac.uk

# Mandibular musculature constrains brain–endocast disparity between sarcopterygians

T. J. Challands[1], Jason D. Pardo[2] and Alice M. Clement[3]

[1]School of Geosciences, University of Edinburgh, Grant Institute, James Hutton Road, Edinburgh, EH9 3FE, UK
[2]Department of Biological Sciences, University of Calgary, Calgary, Alberta, Canada
[3]College of Science and Engineering, Flinders University, Sturt Road, Bedford Park, 5042, South Australia, Australia

TJC, 0000-0002-7759-8966; JDP, 0000-0002-2665-8893;
AMC, 0000-0003-0380-7347

The transition from water to land by the earliest tetrapods in the Devonian Period is seen as one of the greatest steps in evolution. However, little is understood concerning changes in brain morphology over this transition. Here, we determine the brain–braincase relationship in fishes and basal lissamphibians as a proxy to elucidate the changes that occurred over the fish–tetrapod transition. We investigate six basal extant sarcopterygians spanning coelacanths to salamanders (*Latimeria chalumnae, Neoceratodus, Protopterus aethiopicus, P. dolloi, Cynops, Ambystoma mexicanum*) using micro-CT and MRI and quantify the brain–braincase relationship in these extant taxa. Our results show that regions of lowest brain–endocast disparity are associated with regions of bony reinforcement directly adjacent to masticatory musculature for the mandible except in *Neoceratodus* and *Latimeria*. In *Latimeria* this deviation from the trend can be accounted for by the possession of an intracranial joint and basicranial muscles, whereas in *Neoceratodus* difference is attributed to dermal bones contributing to the overall neurocranial reinforcement. Besides *Neoceratodus* and *Latimeria*, regions of low brain–endocast disparity occur where there is less reinforcement away from high mandibular muscle mass, where the trigeminal nerve complex exits the braincase and where endolymphatic sacs occupy space between the brain and braincase wall. Despite basal tetrapods possessing reduced adductor muscle mass and a different biting mechanism to piscine sarcopterygians, regions of the neurocranium lacking osteological reinforcement in the basal tetrapods *Lethiscus* and *Brachydectes* broadly correspond to regions of high brain–endocast disparity seen in extant taxa.

# 1. Introduction

Sarcopterygians comprise about half of all vertebrate diversity today but the vast majority of these are tetrapods (four-limbed vertebrates and their descendants). By contrast, there are just eight extant species of sarcopterygian fish: two congeneric species of coelacanth (*Latimeria* spp.) and three lungfish genera (*Neoceratodus, Lepidosiren, Protopterus*). *Protopterus* contains four species and together with *Lepidosiren* constitute the lepidosirenid lungfishes, which are thought to have diverged from the Neoceratodontid lineage 277 million years ago [1]. Nevertheless, during the Devonian Period (359–419 Ma) sarcopterygian fish were far more diverse and abundant with several now extinct groups, including 'tetrapodomorphs' or tetrapod-like fishes, known [2].

The transition from water to land by some of these sarcopterygians, our earliest tetrapod ancestors, has long been heralded as one of the 'greatest steps' in evolutionary history. The adaptations required in the post-cranial skeleton for this transition to occur, in particular the pectoral limb, have garnered much attention [3–6]. However, it has been long understood that the water–land transition drove sensory adaptations in auditory [7], visual [8] and chemosensory [9] systems. Given these changes in sensory system evolution, it seems likely that these changes to sensory systems would have been accompanied by changes in the central nervous system, including shifts in overall brain size as well as modifications in size and shape of structures of the brain dedicated to the processing of these sensory inputs.

Fossils are an invaluable source of information for uncovering evolutionary history through deep geological time but due to the paucity of soft tissue preservation workers must often rely on osteological correlates to infer soft tissue characteristics of extinct taxa. Palaeoneurology, the study of fossil brains and neural evolution [10], is one such discipline that relies on such methods. Moulds of the internal osseous cavities of the braincase, 'endocasts', are studied in lieu of preserved brains to provide insight into neural evolution. The widespread adoption of modern scanning technologies in palaeontology, especially micro-CT, is currently driving intensified interest in this field [11].

Generally, birds and mammals tend to have larger brains relative to their body mass and a closer correspondence between the brain and braincase, a factor likely to have influenced the dearth of research into the brain–braincase spatial relationships of 'lower' vertebrates by comparison. However, this relationship varies considerably across vertebrates and considerable overlap exists among different groups [12]. For example, some chondrichthyans have a brain that almost completely fills their braincase and a brain–body ratio similar to that in some primates [13], while others have relatively small brains within hugely spacious braincases [14].

While there has been much effort to investigate endocast shape and variation among vertebrates, popularly using geometric morphometrics or outline shape analysis, quantitative analyses of the brain–braincase relationship have remained relatively rare. Just a few studies have recently attempted to determine the reliability of endocasts as proxies of brain size in some non-mammalian taxa [15,16]. A detailed understanding of the relationship between brain and endocast is extremely important in order to avoid erroneous interpretations of physiology and/or behaviour based on the endosseous representations of functional units alone.

Two recent studies highlight the disparity of the brain–braincase relationship in sarcopterygian fishes. Dutel *et al.* [17] confirmed that the brain of the coelacanth *Latimeria* is indeed disproportionately small compared with the braincase housing it. By contrast, Clement *et al.* [15] disproved the assertion that the lungfish brains only occupied approximately 10% of their cranial cavities [18], instead of finding a value closer to 80% for a young individual in their study.

By contrast, there has been relatively little work characterizing brain and endocast shape and size in early tetrapods. Cranial endocasts have been produced for a small number of early tetrapods, most of which belong to the Temnospondyli, a group typically considered to represent the amphibian total group [19]. Endocasts have been produced for the Permian temnospondyls *Eryops megalocephalus* [20,21] and *Edops craigi* [21] and for a handful of Triassic stereospondyls [22] using traditional methods that require the partial or complete destruction of specimens. The advent of modern three-dimensional radiology methods such as X-ray micro-computed tomography (μCT) and propagation phase-contrast synchrotron micro-computed tomography (PPC-SRμCT) has enabled insight into internal braincase anatomy in additional taxa, including the classic Devonian tetrapod *Ichthyostega* sp. [23], the early Carboniferous stem tetrapod *Lethiscus stocki* [24] and the early Permian recumbirostran *Brachydectes newberryi* [25]. Romer & Edinger [21] have drawn limited comparisons with the caudates *Necturus* and *Cryptobranchus* and the anuran *Rana*, noting a relatively 'fish-like' organization of the brain of *Edops*

with a small telencephalon, enlarged hypophyseal canal, and possibly separate tract for the profundus nerve, whereas the endocast of *Eryops* is considerably more 'amphibian-like'.

Moreover, the role that the cranial cavity has in protecting the brain in relation to the surrounding neurocranium and adjacent muscles has not been considered before. The masticatory muscles exert considerable stress on the skull and jaw units to which they are attached [26–29]. From such work, it has become apparent that the morphology of the skull is strongly dependent on the inherent stresses it experiences from muscles. That said, detailed features of the skull such as fenestrae for weight reduction, protection of the brain and sense organs and armour have been dismissed as secondary functions of cranial bone in the context of a cranial morphology that is capable of withstanding the necessary forces exerted by craniofacial muscles [29,30]. These studies, however, were predominantly conducted on mammals and reptiles, and whereas this is likely to be the case for fish as well, arrangement and form of fish cranial bones as a consequence of musculature force distribution is much more poorly known but see Cooper *et al.* as an example of such a study in fishes [31].

Thus, we herein investigate six basal sarcopterygians (*Latimeria, Neoceratodus, P. aethiopicus, Protopterus dolloi, Ambystoma* and *Cynops*) via MRI and micro-CT as a proxy for quantifying the brain–braincase relationship across the fish–tetrapod transition. These data also allow the relationship between brain–endocast disparity to be assessed alongside the neurocranium structure and its muscle distribution, allowing us to compare the system seen in basal tetrapods from the Carboniferous. This work will inform interpretation of sarcopterygian endocasts and shed light on neural evolution in a functional context during the 'greatest step' in evolution.

# 2. Methods

## 2.1. Specimens

### 2.1.1. Coelacanth

The adult specimen, housed in the Collections de Pièces anatomiques en Fluides at the Muséum national d'Histoire naturelle, Paris, is preserved in formalin (8%). The specimen is a male (MNHN C24 (CCC 27)) caught on 4 August 1961, offshore of Grande Comoro, Comoro Islands.

### 2.1.2. Lungfish

The adult lungfish *Neoceratodus forsteri* (SU 18139) was originally collected from Australia in 1980 and then stored in 70% ethyl alcohol in the California Academy of Sciences. The specimen measures 748 mm (standard length) and was rinsed with water thoroughly prior to scanning in 2009. The specimens of *P. aethiopicus* and *P. dolloi* were purchased from a licensed aquarium in the UK and euthanased using an overdose of neutralized 0.05% MS-222 under schedule 1 (UK) at the University of Edinburgh, School of Biological Sciences. Following decapitation, the heads were stained in IKI using the method of [32] for three months.

### 2.1.3. Salamanders

Both specimens were adult individuals acquired through the pet trade. *Cynops* was acquired post-mortem but *Ambystoma* was collected under University of Calgary Animal Care Committee protocol AC15-0020. *Ambystoma* was euthanized in 5% weight by volume MS-222 until heart activity had completely stopped (approx. 1 h). In both cases, the peritoneal cavity was opened, and the animal was fixed in 10% neutral-buffered formalin (NBF) for 24 h and washed in water prior to contrast staining. *Cynops* was stored in 70% ethanol prior to staining, but was rehydrated before being added to the Lugol's iodine solution.

## 2.2. Imaging and segmentation

### 2.2.1. Coelacanth

Specimen MNHN C24 (CCC 27) was immersed for 1 month in a solution of phosphomolybdic acid solution (5% in 70% ethanol) to increase the contrast of the soft tissues for scanning. The specimen was scanned at the AST-RX facility of the Muséum national d'Histoire naturelle, Paris, France, using

the following scanning parameters: voltage, 200 kV; current, 200 µA; number of slices, 1289; voxel size, 0.260 mm; field of view, 262.34 mm. Segmentation was performed using MIMICS v. 15 (Materialise MedicalCo Belgium). The dataset for this specimen is available on request from Dutel *et al.* [17].

### 2.2.2. Australian lungfish

The scan of the adult *Neoceratodus* specimen was obtained using the 3T GE Signa Exite HDx human MRI scanner (Signa Excite 750; GE Healthcare; Milwaukee, WI), equipped with an 8-Channel Cardiac Coil, as part of the Digital Fish Library project [33]. The pulse sequence parameters used were: flip angle of 35 degrees, 12.4 ms repetition time, 3.9 ms echo time and three averages. Images were collected with slice thickness of 0.7 mm and resulting image resolution of 703 microns. Data were converted to DICOM format for image processing and visualization. The magnetic resonance imaging (MRI) scan of the adult *Neoceratodus* specimen was obtained from the Digital Fish Library catalogue, University of California San Diego (http://www.digitalfishlibrary.org). Image segmentation of the brain and braincase was performed using MIMICS v. 17 (Materialise MedicalCo Belgium).

### 2.2.3. *Protopterus aethiopicus* and *Protopterus dolloi*

Micro-CT (µCT) scans were obtained using a Feinfocus 10–160 kV transmission X-ray source and a Perkin Elmer XRD0822 1 MP flat panel X-ray camera. The peak energy used was 120 kV, and 2 s exposures were corrected with offset and gain images. The resulting exposures were reconstructed as a tif stack using OctopusV9 resulting in a voxel size of 87.3 and 86.9 µm for *P. aethiopicus* and *P. dolloi*, respectively. The resulting data were rendered in Drishti v. 2.6.3 and segmented using Drishti Paint [34].

### 2.2.4. Salamanders

DiceCT contrast staining was completed using a Lugol's iodine (I2KI) solution. An 11.75% Lugol's iodine was prepared following the protocol of [35]. *Ambystoma* was stained for 5 days prior to µCT scanning, whereas *Cynops* was scanned after only 72 h. Both specimens were imaged using a Skyscan1173 (Bruker) desktop µCT scanner and were reconstructed as a stack of jpegs using NRecon 1.6.6. In both cases, the files were cropped in ImageJ [36] to reduce file size and sampled images from the stack at an interval of two in the Z-plane. The resultant voxel size for *Ambystoma* is $16.69 \times 16.69 \times 33.38$ µm, and *Cynops* is $15.868 \times 15.868 \times 31.727$ µm. Image segmentation of the brain and braincase was performed using MIMICS v. 17 (Materialise MedicalCo Belgium).

## 2.3. Surface distance analysis and visualization

Spatial overlap and surface distance between the brain and braincase (and its internal 'endocast') were analysed following the methods of Clement *et al.* [15], whereby three-dimensional surface mesh (STL) representations of the brain and endocast were superimposed using iterative closest point (ICP) registration, symmetric mean and maximum absolute distances were computed, and the spatial overlap between the meshes was quantified using a Dice similarity coefficient. For further details, please refer to Clement *et al.* [15] and the electronic supplementary material for the updated script.

## 2.4. Anatomical cross-sectional area of mandibular adductor jaw muscles

The anatomical cross-sectional area of mandibular adductor jaw muscles represents the area of a muscle perpendicular to its longitudinal axis. This was measured using micro-CT and MRI image slices in regions corresponding to sections in figures 1–3 using ImageJ. Ten measurements were taken from five slices for each region and the mean used table 1.

# 3. Results

The segmented brain, endocast, the brain–endocast overlay and disparity distribution between the brain and endocast in all taxa studied herein are shown in figures 4–7. Although figure 7 shows absolute distances (in mm), interspecific differences may be described in relative terms, as all specimens represent adults. For example, even though the maximum distance between the brain and endocast in

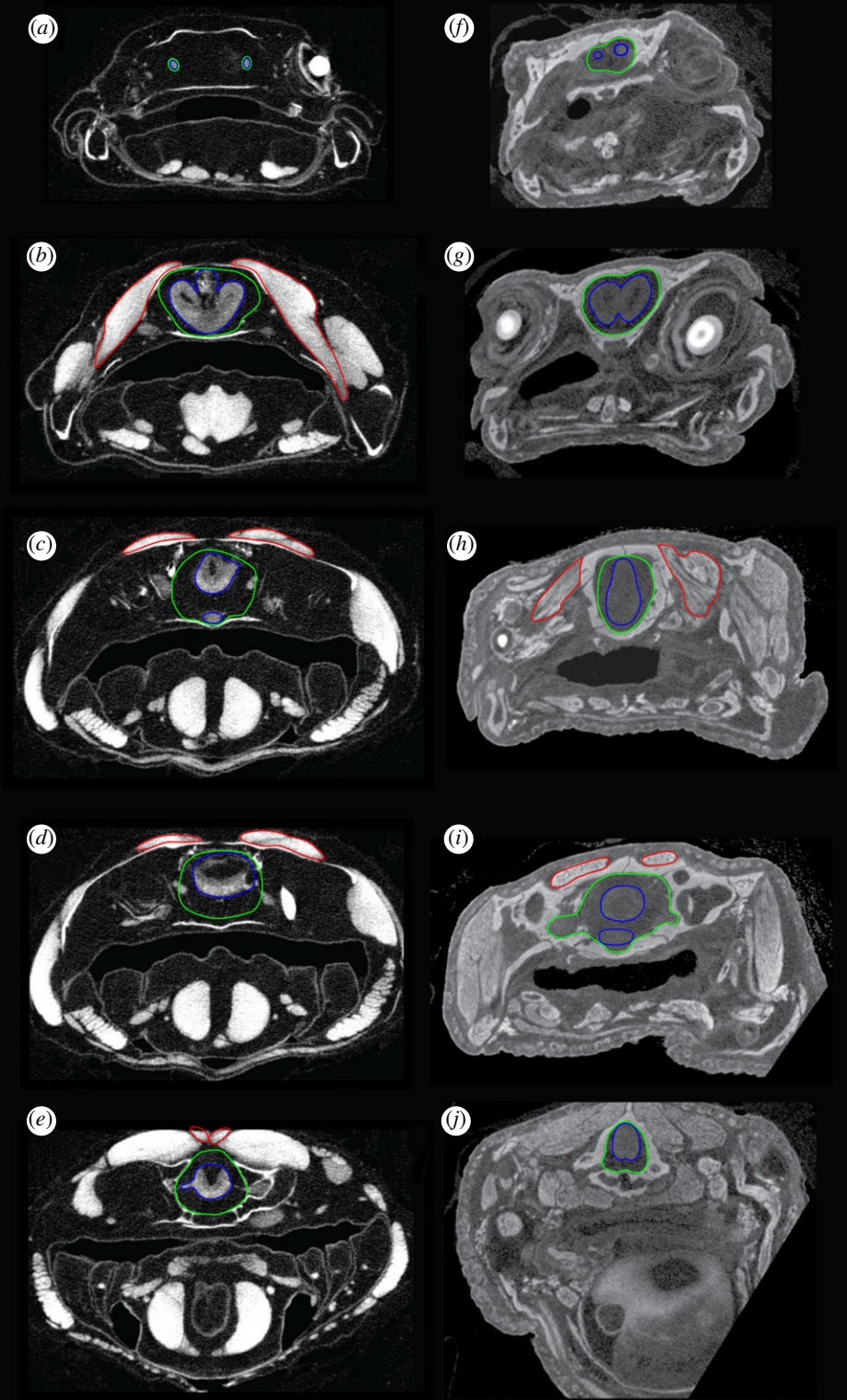

**Figure 1.** DiceCT sections through the olfactory tracts (*a* and *f*), telencephalon (*b* and *g*), diencephalon (*c* and *h*), region of the trigeminal nerve complex (*d* and *i*) and region of the vagus nerve (*e* and *j*) in *Ambystoma* (left-hand column) and *Cynops* (right-hand column). The brain is outlined in blue, the cranial endocast in green and the adductor mandibularis muscle complex in red.

*Latimeria* and *Ambystoma* is 179.0 and 1.25 mm, respectively, it is more meaningful to speak in terms of where the maximum distances are topographically, because adults of both taxa are not the same size.

Several distinct observations are apparent from the data comparing disparity between the brain and the cranial endocast: (i) *Latimeria* is anomalous in its brain–endocast volume ratio compared with other

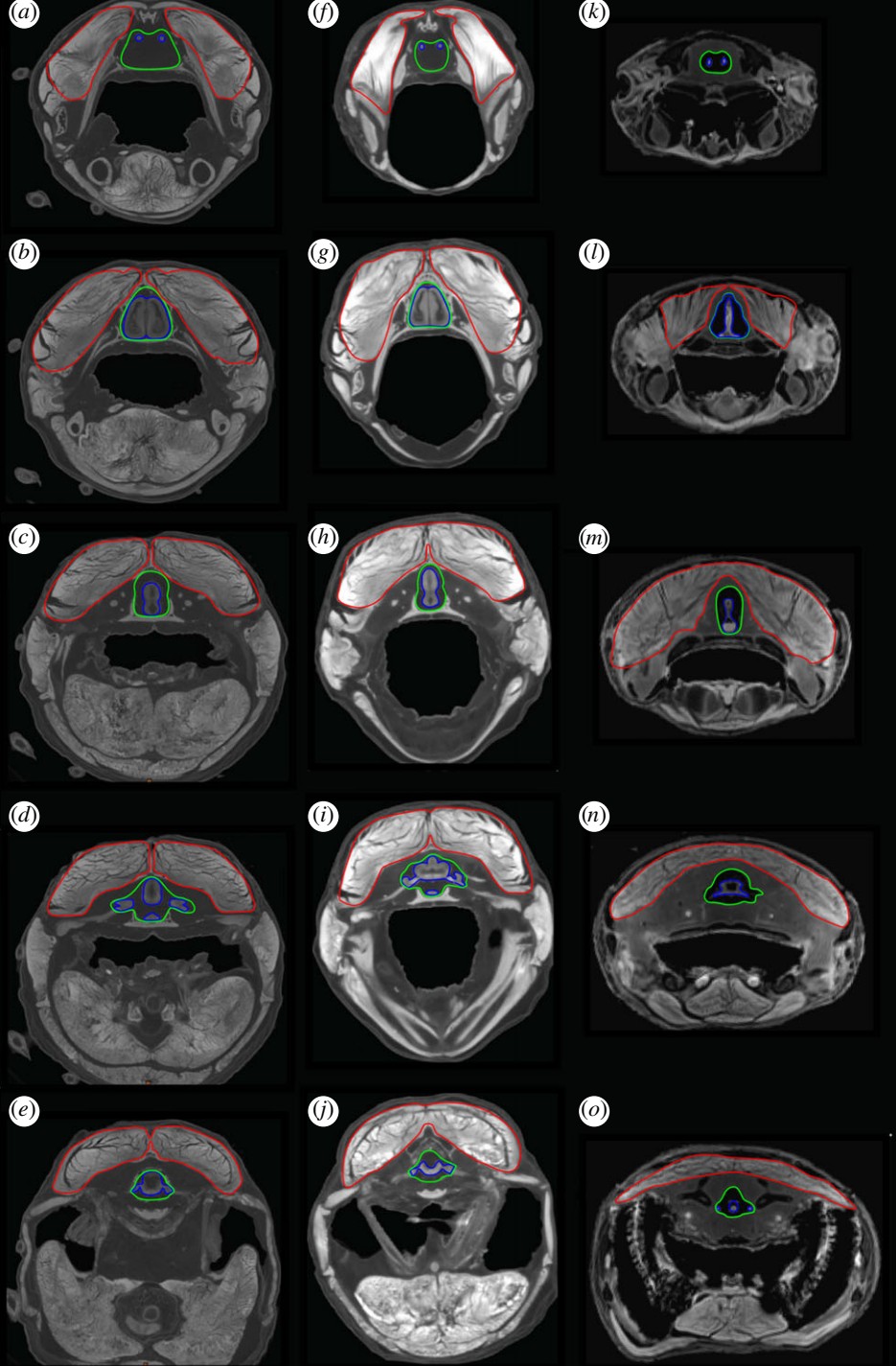

**Figure 2.** DiceCT sections through the olfactory tracts (*a,f* and *k*), telencephalon (*b,g* and *l*), diencephalon (*c,h* and *m*), region of the trigeminal nerve complex (*d,i* and *n*) and region of the vagus nerve (*e,j* and *o*) in *P. aethiopicus* (left-hand column), *P. dolloi* (centre column) and *Neoceratodus* (right-hand column). The brain is outlined in blue, the cranial endocast in green and the adductor mandibularis muscle complex in red.

extant sarcopterygians; (ii) *Neoceratodus* differs considerably in both brain and endocast morphology from lepidosirenid lungfishes; (iii) the endocasts of the lepidosirend lungfishes, *Ambystoma* and *Cynops* do not exhibit a distinct hypophyseal fossa; (iv) the region anterior ventral to the telencephalon and the regions corresponding to the trigeminal complex of the extant lungfish endocasts studied herein show the greatest brain–endocast disparity, whereas the telencephalon in the tetrapods has a tighter fit anteriorly. Instead, the region of greatest disparity between endocast and brain in the salamanders is

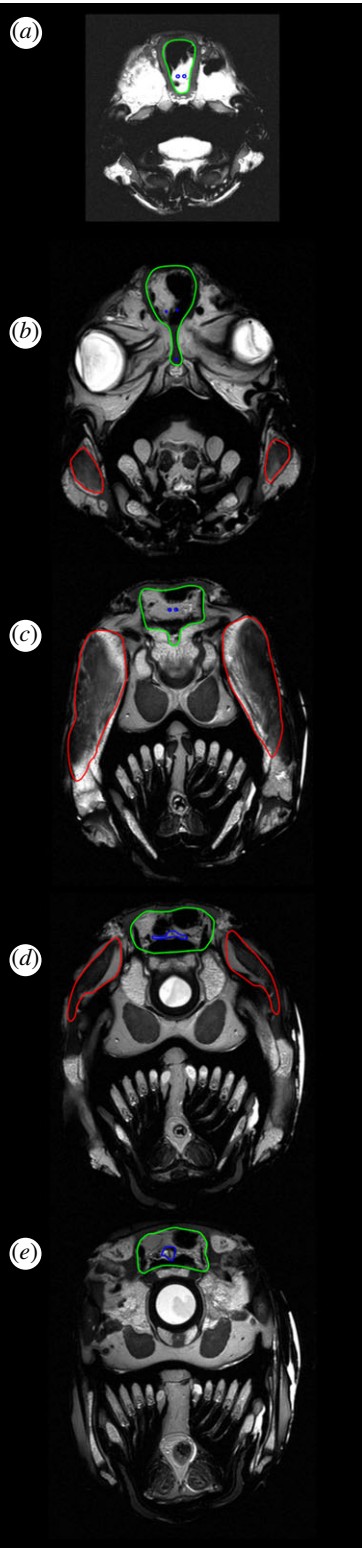

**Figure 3.** DiceCT sections through the olfactory tracts (*a*), telencephalon (*b*), diencephalon (*c*), region of the trigeminal nerve complex (*d*) and region of the vagus nerve (*e*) in *Latimeria*. The brain is outlined in blue, the cranial endocast in green and the adductor mandibularis muscle complex in red.

the mesencephalon; and (v) the region of greatest masticatory muscle mass corresponds to the region of tightest brain–endocast fit in all taxa except *Neoceratodus* and *Latimeria*.

The cranial endocast of *Ambystoma mexicanum* in the specimen studied measures approximately 16.6 mm in length, 4.8 mm wide and 3.8 mm deep at its deepest point. From lateral view, it appears

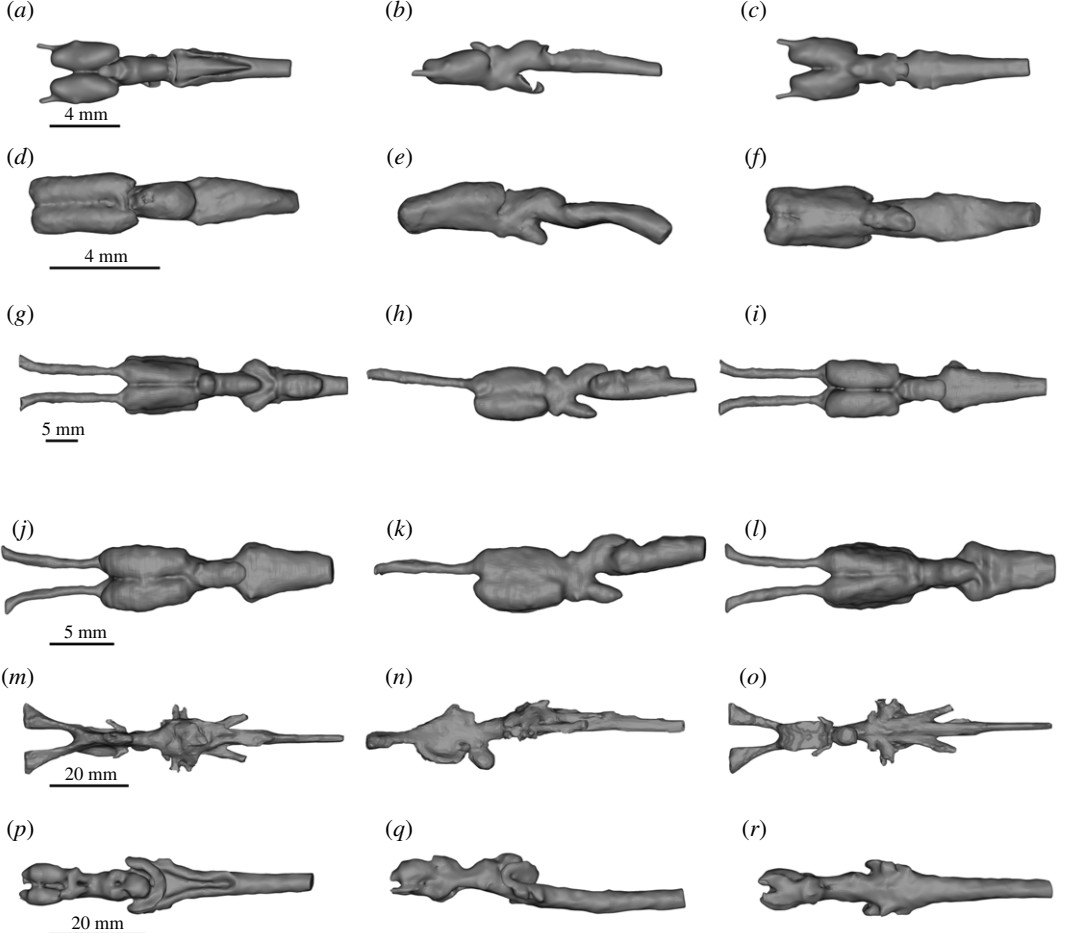

**Figure 4.** DiceCT (*a–l*) and microMRI (*m–r*) renderings of the brain in dorsal (left column, *a,d,g,j,m,p*), lateral (middle column, *b,e, h,k,n,q*) and ventral (right column, *c,f,i,l,o,r*) of extant piscine sarcopterygians and salamandrids. (*a–c*) *A. mexicanum*; (*d–f*) *Cynops*; (*g–i*) *P. aethiopicus*; (*j–l*) *P. dolloi*; (*m–o*) *N. forsteri*; (*p–r*) *Latimeria chalumnae*.

**Table 1.** Brain volume as a percentage of endocast volume in mm$^3$.

| taxon | brain volume (mm$^3$) | endocast volume (mm$^3$) | % |
|---|---|---|---|
| *Ambystoma* | 27.5 | 73.7 | 37 |
| *Cynops* | 14.0 | 34.4 | 41 |
| *Latimeria* | 1972.9 | 201 276.0 | 1 |
| *Neoceratodus* | 6298.0 | 14 027.0 | 45 |
| *P. aethiopicus* | 456.7 | 1096.5 | 42 |
| *P. dolloi* | 407.2 | 857.4 | 47 |

approximately straight, whereas from a dorsal perspective, it tapers laterally from the anterior to the posterior. Canals for the olfactory tracts are short (approx. 1.25 mm) before they enter the olfactory epithelium. The depth of the endocast decreases abruptly approximately half-way along corresponding to the optic tectum dorsally and posterior to the hypophysis ventrally. There is no ventrally directed hypophyseal recess. The mandibular adductor muscle complex does not extend anteriorly as far as the olfactory tracts and exhibits its greatest muscle mass area (4.78 mm$^2$) around the telencephalon (figure 8). At the diencephalon, the trigeminal region, and the region of the vagus nerve (nX)—the regions corresponding to increased brain–endocast disparity—the mandibular adductor muscle cross-sectional area decreases abruptly to 0.75, 0.56 and 0.15 mm$^2$, respectively.

**Figure 5.** DiceCT (*a–l*) and microMRI (*m–r*) renderings of the cranial endocasts in dorsal (left column, *a,d,g,j,m,p*), lateral (middle column, *b,e,h,k,n,q*) and ventral (right column, *c,f,i,l,o,r*) of extant piscine sarcopterygians and salamandrids. (*a–c*) *A. mexicanum*; (*d–f*) *Cynops*; (*g–i*) *P. aethiopicus*; (*j–l*) *P. dolloi*; (*m–o*) *N. forsteri*; (*p–r*) *L. chalumnae*.

The cranial endocast of *Cynops* differs from *Ambystoma* in being widest at the region of the trigeminal complex (3.9 mm). This corresponds to the region of greatest disparity (1.15 mm) between the brain and the cranial endocast in *Cynops*. The total length of the endocast is 10.3 mm, and it is 2.5 mm deep at its thickest point which, like *Ambystoma*, is in the region of the optic tectum and hypophysis. The hypophysis of *Cynops* is directed posteriorly, and there is no ventrally directed hypophyseal recess. Unlike *Ambystoma,* where the rhombencephalon of the brain and the anterior spinal cord are straight, the corresponding regions in *Cynops* curve dorsoventrally. This curvature is also reflected in the endocast. Similarly, the ventral region of the telencephalon is slightly curved with the anterior region and olfactory bulbs appearing to point ventrally. This curvature produces a slightly greater discrepancy between brain and endocast in this region than in *Ambystoma*. The cranial endocast of *Cynops* is constricted in the region of the diencephalon but then expands again to the posterior where the trigeminal nerves exit the brain. This produces a cavity in the region lateral to the mesencephalon. In *Ambystoma*, this constriction is not readily apparent in the endocast but is marked by the endocast starting to taper. As such, *Ambystoma* has a greater discrepancy between brain and endocast in this region.

The distribution of muscle mass relative to the endocast in *Cynops* is similar to that in *Ambystoma* in the posterior region though differs by the mandibular adductor muscle complex not extending as far anteriorly as the telencephalon. The greatest cross-sectional area (3.41 mm$^2$) is seen in the region of the

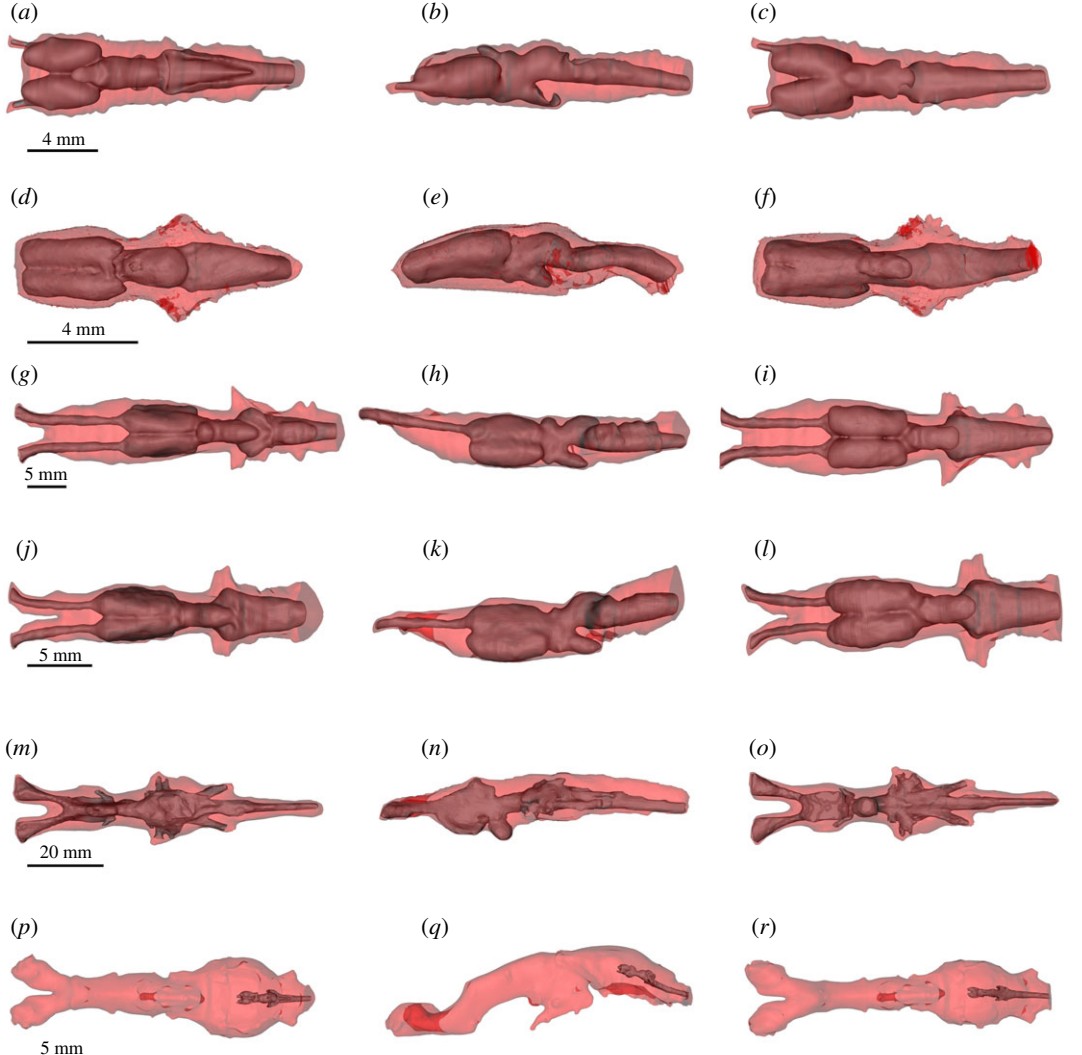

**Figure 6.** DiceCT (*a–l*) and microMRI (*m–r*) renderings showing the position of the brain (grey) in the cranial endocast (transparent red) in dorsal (left column, *a,d,g,j,m,p*), lateral (middle column, *b,e,h,k,n,q*) and ventral (right column, *c,f,i,l,o,r*) of extant piscine sarcopterygians and salamandrids. (*a–c*) A. mexicanum; (*d–f*) Cynops; (*g–i*) P. aethiopicus; (*j–l*) P. dolloi; (*m–o*) N. forsteri; (*p–r*) L. chalumnae.

diencephalon where it is directed ventrally to attach to the mandible. In *Cynops*, this region exhibits low disparity between the brain and endocast. Posterior to the diencephalon the muscle cross-sectional area displays a similar trend to *Ambystoma* and decreases (trigeminal region = 0.76 mm$^2$, vagus region = 0.17 mm$^2$) where the muscles are routed and attached to the parietal.

In *P. aethiopicus*, the cranial endocast measures 41 mm from the point at which the olfactory tracts enter the olfactory epithelium to the anteriormost 4 mm of the spinal cord. The anteriormost portion of the endocast exhibits a canal 3.7 mm in length housing diverging olfactory tracts. This represents only 28% of the total length of the olfactory tracts which are 13.1 mm long. Overall the endocast is long and broad, noticeably so in the telencephalon region and the rhombencephalon and spinal cord regions. The dorsal surface is flat with the canal for the olfactory tracts lying in the same plane as the dorsal surface.

In *P. aethiopicus*, the mandibular adductor muscle complex extends anteriorly to lie adjacent to the olfactory tracts. It is in this region that the muscle descends ventrally to attach to the prearticular, but while it lies in close proximity to the region of high brain–endocast disparity, the cross-sectional area of the muscle here is low (17.31 mm$^2$). The muscle is also separated laterally from the cranial cavity and brain by the pterygoid. In the telencephalon region, the adductor muscle cross-sectional area is at its greatest (19.67 mm$^2$). The cranial cavity in this region is supported by a ventral projection of the frontoparietal as well as a dorsal expansion of the parasphenoid. Where the disparity between brain

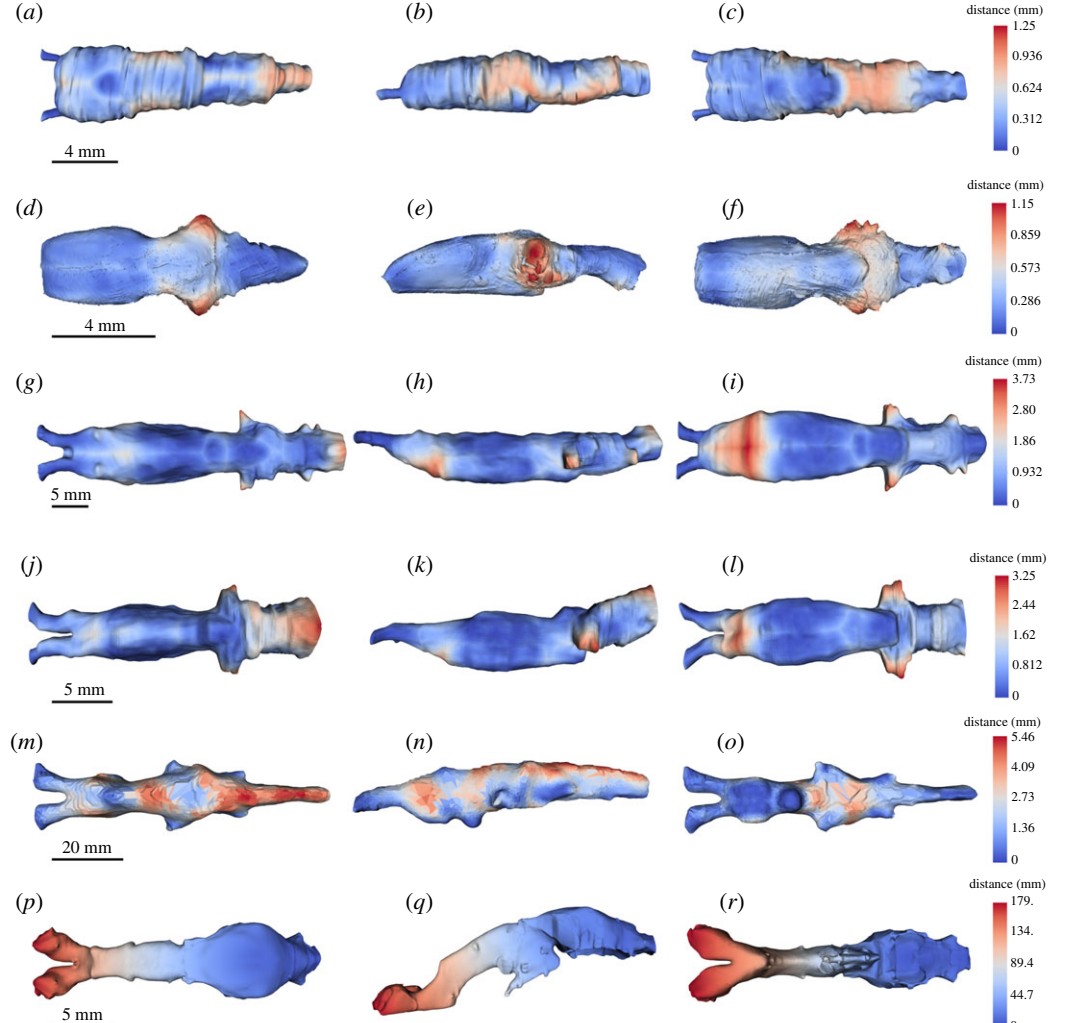

**Figure 7.** Unsigned distance (mm) from the brain to cranial endocast superimposed on renderings of the cranial endocast in dorsal (left column, *a,d,g,j,m,p*), lateral (middle column, *b,e,h,k,n,q*) and ventral (right column, *c,f,i,l,o,r*) of extant piscine sarcopterygians and salamandrids. (*a–c*) *A. mexicanum*; (*d–f*) *Cynops*; (*g–i*) *P. aethiopicus*; (*j–l*) *P. dolloi*; (*m–o*) *N. forsteri*; (*p–r*) *L. chalumnae*.

and endocast is greater in the diencephalon, trigeminal nerve region and the region of the vagus nerve, the cross-sectional area of the mandibular muscle mass decreases to 101.47, 89.49 and 68.45 mm$^2$ respectively.

The same pattern is seen in *P. dolloi* where the mandibular muscle cross-sectional area is lowest in the region of the olfactory tracts (14.54 mm$^2$), greatest in the telecephalon region (111.03 mm$^2$) and then decreases towards the region of the vagus nerve (101.47, 89.49 and 68.45 mm$^2$ for the diencephalon, trigeminal region and vagus nerve regions respectively).

The cranial endocast of a juvenile *Neoceratodus* has been described previously by Clement *et al.* [15], so in this description, we only highlight differences between the morphology in their study to that seen here. Clement *et al.* [15] incorporated the labyrinth system in their brain and endocast reconstructions. In our analysis, the labyrinth has not been included as it is a different functional system, though it may be noted that in some taxa the total cranial endocast does include the labyrinth system (e.g. *Neoceratodus*), while in others (e.g. *Protopterus* and *Latimeria*), the two are clearly separated. Where the labyrinth and cranial cavity are incorporated, the extent of the endocast of the cranial cavity that houses the brain can be determined from the presence of stained soft tissues.

The cranial endocast of the specimen of *Neoceratodus* used in this study is 91.3 mm long and 23.1 mm at its widest point—the region where the trigeminal nerves exit the brain. The deepest point of the endocast of *Neoceratodus* is in the region of the diencephalon where a distinct hypophyseal recess results in a depth of 17.2 mm. In the endocast of Clement *et al.* [15], the deepest point occurs in the

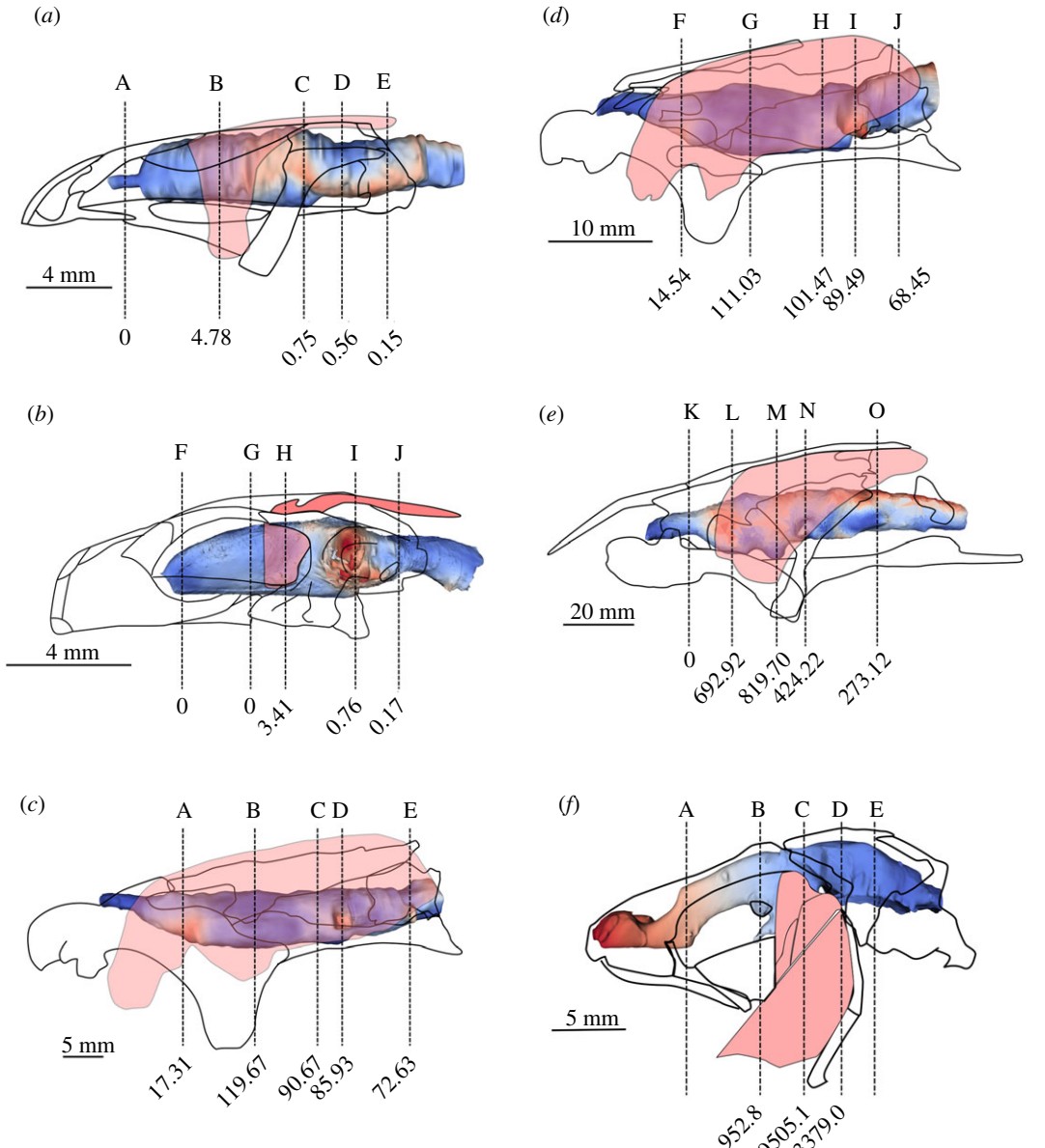

**Figure 8.** Unsigned distance map of brain to cranial endocast distance shown in position relative to the neurocranium, skull and mandibular adductor complex in (a) *Ambystoma*; (b) *Cynops*; (c) *P. aethiopicus*; (d) *P. dolloi*; (e) *N. forsteri*; (f) *L. chalumnae*. Lettered vertical lines indicate respective sections in figure 1 for *Ambystoma* and *Cynops*, figure 2 for *P. aethiopicus*, *P. dolloi* and *Neoceratodus* and figure 3 for *Latimeria*. Values at the base of vertical sections indicate the anatomical cross-sectional area (mm$^2$) for the adductor mandibularis muscle complex at that section.

labyrinth. The canal for the olfactory peduncles is longer in the adult specimen used in this study and exhibits a greater disparity between brain and endocast in this region than in the juvenile specimen.

*Neoceratodus* displays a different pattern of the distribution of the mandibular adductor muscle cross-sectional area compared with the salamanders and *Protopterus*. In *Neoceratodus*, the mandibular adductors do not extend far anteriorly, such that there is no muscle adjacent to the olfactory tracts. The most anterior extent of the mandibular adductors is adjacent to the telencephalon where it passes over the pterygoid ramus which, in *Neoceratodus*, is not as ventrally depressed as in *Proptopterus*. At this point, the adductor muscle cross-sectional area is 692.92 mm$^2$. However, the greatest cross-sectional muscle area (819.70 mm$^2$) is seen in the region of the diencephalon in *Neoceratodus* which, conversely, also corresponds to a region of increased brain–endocast disparity, particularly in the dorsal region. In this region, the brain and cranial cavity are further reinforced laterally by the squamosal. Posterior to this, the pattern of muscle cross-sectional area and brain–endocast disparity is similar to that in *Protopterus* with the muscle area decreasing to 424.22 and 273.12 mm$^2$ in the regions of the trigeminal and vagus nerves, respectively.

The olfactory tracts of *P. aethiopicus* and *P. dolloi* are considerably longer than those in *N. forsteri* (figure 4*g–o*, figure 6*g–o*), and whereas the former possess sessile olfactory bulbs, *Neoceratodus* possesses pedunculate olfactory bulbs positioned at the end of olfactory peduncles, which should not be confused with olfactory tracts which are positioned anterior to the olfactory bulb. Clement *et al.* [15] noted that the olfactory bulbs of juvenile *Neoceratodus* possessed shorter olfactory peduncles relative to adult specimens, but in both cases, there is only a small amount of space between the olfactory tracts and the endocast compared with that seen in both species of *Protopterus* (figure 6).

The specimen of *Neoceratodus* used in this study exhibits high disparity between brain and endocast in the lateral regions of the telencephalon which is not seen in the juvenile *Neoceratodus* or *Protopterus*. In this instance, this can be attributed to shrinkage of the brain in the *Neoceratodus* specimen following storage in ethyl alcohol for 40 years. Other regions of the brain that show high disparity (the dorsal and ventral regions of the mesencephalon; the dorsal region of the rhombencephalon and spinal cord) are consistent with observations of Clement *et al.* [15].

The spinal cord region of both *Neoceratodus* and *P. dolloi* exhibit a large difference between the endocast and the spinal cord. Of note is that the endocast of this region in *P. dolloi* expands considerably posterior to the labyrinth region. This is seen to a lesser extent in *P. aethiopicus* and functions to accommodate the vagus nerve (nX). In the endocast of *Neoceratodus* and *Protopterus*, the vagus nerve emanates from the region corresponding to the rhombencephalon immediately posterior to the labyrinth, but in the endocast of *P. dolloi*, this nerve is housed in a cavity formed by the adjacent cleithra.

The brain–endocast relationship of *Latimeria,* including ontogenetic changes, was described in detail by Dutel *et al.* [17] who confirmed the extreme difference between the brain and endocast size and morphology in adults. Our results determine the region of greatest disparity in an adult specimen to be in the extreme anterior portion of the canals for the olfactory peduncles, with the brain itself being restricted to the portion of the endocasts posterior to the intracranial joint. Posterior to the intracranial joint, the distance between brain and endocast is somewhat consistent. The large distances between the spinal cord and the corresponding endocast region seen in the lungfish and tetrapods are not evident nor is the increased space seen in the region where the trigeminal nerves exit the brain. In some respects, the endocast of *Latimeria* most closely resembles that of the lepidosirenid lungfishes which exhibit an anterior region separated from the brain. However, in the lepidosirenid lungfishes, this region shows the disparity between the endocast and long olfactory tracts rather than olfactory peduncles as seen in *Latimeria*. The similarity must therefore only be considered as analogous rather than homologous.

# 4. Discussion

## 4.1. The intracranial joint

*Latimeria* is unique among those taxa analysed in this study in being the only one with an intracranial joint. The extreme reduction in brain size in *Latimeria* is probably related to the enlargement of the notochord and biomechanical constraints related to the persistence of the intracranial joint [17]. The similarities in the increase in space in the region anterior to the telencephalon seen in *Protopterus*, as mentioned, cannot be considered in the same light due to the fusion of the intracranial joint in lungfish. Without an ontogenetic sequence in modern lungfish, it is not possible to determine the extent to which growth of the notochord influences the development of the brain and the endocranial cavity.

However, in *Latimeria*, the adductor mandibularis all lie just anterior to the intracranial joint which, in the adult *Latimeria* endocast, is a region of high brain–endocast disparity (figures 7 and 8). Rather than this configuration jeopardizing the integrity if the neurocranium and its contents, the external force exerted in the neurocranium in this region is probably limited by the presence of the palatoquadrate and a basicranial muscle that increases bite force in *Latimeria*. The basicranial musculature is not seen in lungfish or tetrapods as a consequence of the loss of the intracranial joint, and in extant sarcopterygians it is unique to coelacanths. Furthermore, the cartilaginous portion of the neurocranium itself is actually supported in this region of musculature by the basisphenoid, the parasphenoid and the pterygoid [37].

Whereas *Latimeria* is the only extant sarcopterygian in possession of an intracranial joint, it was ubiquitous in all basal sarcopterygians, including tetrapodomorphs but independently lost in lungfish and tetrapods and occurred alongside a fusion of the palatoquadrate with the neurocranium. The mechanical constraints of the neurocranium in tetrapodomorphs and hence the brain–endocast

relationship may at first be considered to be similar to that of *Latimeria*. However, the pattern of ossification in finned tetrapodomorphs was considerably different from that of *Latimeria* with the entire neurocranium being ossified (rhizodonts being an exception). This undoubtedly provided greater protection to the brain from stresses created during biting, though the degree to which this would have been necessary is uncertain. However, complete ossification of the braincase appears to have been lost very early in tetrapods; ossification of the ethmoid and nasal capsules is lost within the fin-to-limb transition [38], and ossification of the sphenoid and metoptic region is lost in diverse lineages [38,39]. It is therefore unclear how the neurocranium of early tetrapods might have adapted to, or facilitated, shifts in bite force capacity during terrestrialization. Interestingly, there is little evidence that early tetrapods exhibited substantial changes in biomechanical properties of jaw closure across the fin-to-limb transition [40,41].

Modelling of the lower jaw morphology, stress distribution and relative bite force in tetrapodomorphs demonstrate no clear correspondence between relative bite force and possession or lack of an intracranial joint [41]. Relative bite force and von Mises stress both remain relatively low following the loss of the intracranial joint and through the fin-to-limb transition. This is consistent with prior studies showing that the evolution of novel biomechanical capability in the lower jaw occurred relatively late in tetrapod evolution and was largely restricted to the tetrapod crown group [40].

However, these studies were based on the mandible alone and were not able to consider the total mandibular adductor muscle force on the mandible or the neurocranium. In extant salamanders Fortuny *et al*. [42] found that the effect of bite force on the neurocranium from the mandibular adductor muscles in the Chinese giant salamander *Andrias davidanus* was very low relative to the dermal skeleton and the jaw components. They found that the regions of highest stress associated with use of the mandibular adductor muscle complex were confined to the pterygoid, maxilla and frontals in the anterior of the skull and in the quadrate and squamosal in the posterior region of the skull. This suggests that the arrangement of dermal bones provide a sufficient protective system for the neurocranium in their own right with the neurocranium acting as a scaffold on which the stress-bearing elements are hung. Conversely, estimations of stress distribution in Triassic stereospondyl amphibians demonstrate high-stress regimes in the posterior braincase and parasphenoid [43]. The dermal skull of Triassic stereospondyl amphibians, however, are considerably more robust than those of many Carboniferous tetrapods and, indeed, the salamandrids and so may be poor analogues for Carboniferous tetrapods. Unfortunately, the cranial endocasts for Devonian and Carboniferous tetrapods are poorly known and the only complete early tetrapod endocast is the Visean (Lower Carboniferous) *Lethiscus stocki* [24].

## 4.2. Early tetrapod endocasts

In this context, we can interpret some trends in brain anatomy in early tetrapods based on published endocast descriptions. The endocast of *L. stocki* has been figured by Pardo *et al*. [24] and represents the first complete endocast of a stem tetrapod close to the fin-to-limb transition. The endocast of *Lethiscus* is small, with an extremely small and narrow cerebellar fossa, although it appears to have some degree of swelling in the midbrain, possibly associated with extrapolation of the optic tectum (figure 9*a*). Although our study here would suggest that endocast structures associated with the midbrain show lower fidelity with overall brain morphology, we note that jaw musculature in *Lethiscus* would have been fully compartmentalized between the palatoquadrate and cheek and thus likely not to have produced compression of the endocranial cavity. Expansion of the optic tectum and associated midbrain structures in *Lethiscus* might lend support to recent hypotheses that the fin-to-limb transition might have included a dramatic shift in visual capacity in the tetrapod lineage [8].

Interestingly, this is similar in some key ways from cranial endocast morphology described for several other Palaeozoic tetrapods. The endocast of the temnospondyl *Edops craigi* has been described by Romer & Edinger [21]. *Edops* is an early-diverging member of the Temnospondyli, a group which is thought to represent the lissamphibian total group [24,44–46]. *Edops* notably differs from both modern salamanders and modern amniotes, but is consistent with the morphology of *Lethiscus* in having a small, narrow telencephalon and expanded midbrain region, with an anteriorly displaced pineal organ situated on a long stalk [21].

Romer & Edinger [21] also identified a small recess dorsomedial to the anterior semicircular canal as an impression of the endolymphatic sac, in a position similar but slightly anterior to that seen in fossil lungfishes [47–50] as well as modern lungfishes (figure 10). Interestingly, the endolymphatic sac in salamanders is expanded ventrally to cover most of the lateral surface at the division between

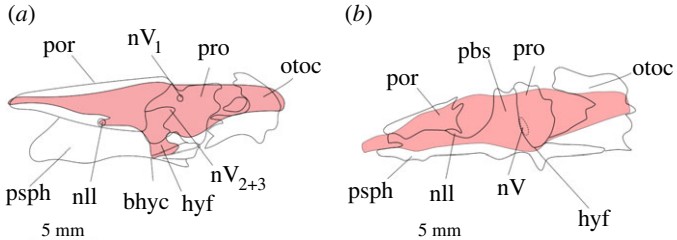

**Figure 9.** Outline of ossified neurocranial elements with endocast superimposed (light red) of (*a*) *Lethiscus* and (*b*) *Brachydectes*. The region of the olfactory tracts is unsupported by skeletal elements while, conversely to extant lungfish and tetrapods, the region of the trigeminal complex is partially enclosed and supported, possibly implying less brain–endocast disparity here. The hypophyseal fossa is oriented posteroventrally in *Lethiscus*. Abbreviations: bhyc, buccohypophyseal canal; hyf, hypophyseal fossa; nll, passage of optic nerve II; nV, passage of trigeminal nerve complex (undifferentiated); $nV_1$, passage of profundus nerve; $nV_{2+3}$ passage of maxillomandibular branch of trigeminal nerve; otoc, ottoccipital; pbs, parabasisphenoid; por, ossification of preoptic root; pro, prootic; psph, parasphenoid.

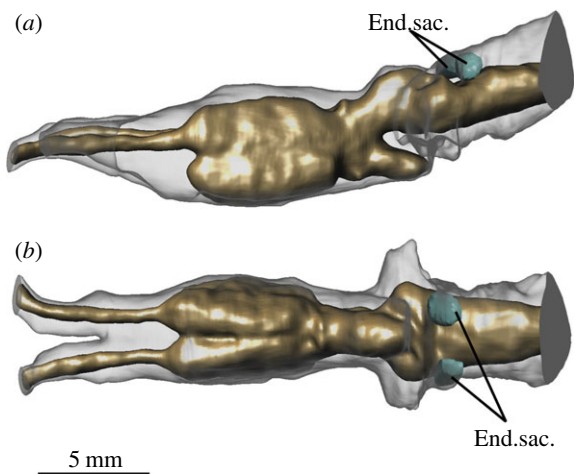

**Figure 10.** Position of endolymphatic sacs accounting for space between posterodorsal region of brain and endocast in *P. dolloi*. (*a*) lateral view; (*b*) dorsal view.

midbrain and hindbrain, corresponding with one of the regions of higher discordance between brain and endocast in the axolotl, but this does not appear to be the case in *Edops*. The condition in *Edops* also differs from that observed in the more advanced temnospondyl *Eryops*, which exhibits a somewhat enlarged telencephalon and distinct dorsal and ventral tracts for the olfactory nerve [20] similar to the condition seen in modern caecilians [43].

An endocast of the recumbirostran 'microsaur' *Brachydectes newberryi* has also been described by Pardo & Anderson [25] using three-dimensional μCT (figure 9*b*). Although sometimes considered to be close relatives of modern amphibians [51,52], recent work has shown a close affinity between recumbirostrans and amniotes [25,45,53]. In *Brachydectes*, the neurocranium is heavily ossified except for the metoptic region, allowing for reconstruction of a rather complete cranial endocast. The endocast is large in comparison with the absolute size of the head, has an expanded space for the cerebral hemispheres and olfactory lobes and a strongly constricted midbrain region. The hypophyseal region is extremely poorly delineated in the endocast, consistent with our findings here that the hypophyseal region may be a region of poor fidelity between brain and endocast. Interestingly, of the early tetrapods for which we have cranial endocasts, *Brachydectes* is the only taxon where the mandibular adductor system would have interacted with the neurocranium and vault. Furthermore, the neurocranium of *Brachydectes* is robustly formed and heavily ossified, possibly to resist compression of the cranial space during head-first burrowing (figure 9*b*). Regardless of this shift in jaw adductor musculature, the endocast of *Brachydectes* shows an organism with a substantially expanded telencephalon in comparison with earlier tetrapod lineages.

## 4.3. Hypophyseal region

The region of greatest disparity between endocast and brain in the salamanders happens to be in the region directly lateral to and posterodorsal to the hypophysis in the mesencephalon region (figures 6 and 7). This may be a consequence of the posterior redirection of the hypophysis forming a conspicuous gap between itself and the ventral margin of the mesencephalon. By contrast, *Neoceratodus* and *Latimeria* have distinct protruding hypophyseal fossae whereas the lepidosirenid lungfish and the extant tetrapods *Ambystoma* and *Cynops* lack this. *Latimeria* and *Neoceratodus* differ in this aspect between each other, with the hypophyseal recess extending anteroventrally and posteroventrally in the latter. Interestingly, the former condition is seen in the endocasts of extinct lungfish [47–50] and may be considered the primitive condition for the Dipnomorpha. The hypophyses of the salamandrids and extant lungfishes are all directed in a posteroventral direction, but their extent and orientation is more difficult to ascertain from the shape of their endocasts alone. The phylogenetic implications of this observation are significant given the condition in *Lethiscus* which also exhibits a posteroventrally oriented hypophyseal fossa. Whereas the polarity of the different types of hypophyseal fossa requires further examination in order to determine their polarity, Lu *et al.* [54] noted that in tetrapodomorphs and onychodonts the hypophyseal fossa extended directly ventrally and lacked a posterior lobe, whereas in the Dipnomorph *Powichthys* and *Latimeria* it extended anteroventrally. The recognition of a posteroventrally extending hypophyseal fossa in the taxa studied herein is different from that of the Devonian coelacanth *Diplocercides* and the Dipnomorph *Youngolepis* which do have a directly ventrally extending fossa but are of the type bearing a posterior lobe. The hypophyseal fossa itself has been described in some detail across early tetrapods [24,38] and is relatively deep in early tetrapods outside of the tetrapod crown [21,24,25,38,39,53]. The endocast of *Lethiscus* may still possess a posteriorly directed lobe but it is clearly also directed posteroventrally from where it emanates from the body of the endocast. That this type of structure is seen in both lungfishes and tetrapods suggests convergence associated with loss of the intracranial joint. At which point, the hypophyseal fossa reoriented in the tetrapodomorph-tetrapod lineage is unknown—tetrapodomorphs basal to *Eusthenopteron* possess an anteroventrally oriented hyposphyseal fossa while in *Eusthenopteron* itself it is oriented ventrally [54].

## 4.4. Lungfish endocasts

The brain–endocast data for *Neoceratodus* presented herein therefore appears misleading in suggesting that, like *Protopterus* spp., it too has a great disparity in the anterior telencephalon and olfactory tract region. This is easily attributed to shrinkage of the brain, having been preserved and stored in alcohol. Such shrinkage has been documented in other taxa treated with iodine staining prior to micro-CT scanning. Critically, Carlisle *et al.* [55] documented shrinkage of slices of mouse brain of up to 45% without prior treatment in hydrogel. In particular, they note that shrinkage was greatest in the olfactory bulb and cerebral cortex region, consistent with the observation here of shrinkage in the telencephalon region in the specimen of *Neoceratodus* used in this study. That said, just as *Latimeria* exhibits a large degree of brain–endocast disparity during ontogeny [17], an alternative explanation could be that difference in brain–endocast volume seen between the juvenile and adult *Neoceratodus* specimens could be a less exaggerated instance of ontogenetic disparity. However, given that the regions of shrinkage are similar to those seen from experimental data [55] and that *Neoceratodus* does not have the same biomechanical constraints imposed by the presence of an intracranial joint, we suggest the brain–endocast relationship seen in juvenile *Neoceratodus* to more closely represent the true proportions of an adult specimen though the ontogenetic effects of brain–endocast relationships are still poorly understood in fish. In addition to possible ontogenetic effects, intraspecific brain–endocast disparity is known to exist between different sexes, during breeding cycles and for different habitats in teleosts [56,57] and also in anuran tetrapods [58,59]. No such patterns are currently known for piscine sarcopterygians, and whereas our data are not able to address the possible contribution of such factors towards brain–endocast disparity, skull structure and jaw musculature, they are undoubtedly valuable avenues for future research to aid our understanding of the piscine sarcopterygian and basal tetrapod brain.

One surprising result of the analysis of this study is that the cranial endocast of *Neoceratodus* differs considerably in both brain and endocast morphology from lepidosirenid lungfishes. It has long been recognized that the brain of *Neoceratodus* differs quite significantly from those of lepidosirenid lungfish and is in fact said to have more features in common with *Latimeria* than *Protopterus* and

*Lepidosiren* [60–62]. For example, the cerebellum is well-developed in *Latimeria* and *Neoceratodus*, whereas it is less distinct in lepidosirenid lungfishes and lissamphibians [61,62]. Despite this, it may have been expected for the cranial endocast to closely resemble that of the brain as it does in *Neoceratodus*. Instead, the endocasts of *Protopterus* spp. are relatively simple and tube-like, with an obvious widening in the telencephalic region.

It is the anterior telencephalon region of the lepidosirenid lungfishes that shows the greatest difference between brain and endocast, whereas the telencephalon in the tetrapods has a tighter fit anteriorly. Fossil lungfish crownward of the Middle Devonian lungfish *Dipterus* exhibit an endocast with an expanded ventral telencephalon [47,48] which in extant taxa is occupied by an enlarged subpallium and sessile olfactory bulbs. This space is not seen in *Neoceratodus* which, despite possessing an expanded subpallium, has pedunculate olfactory bulbs. Pedunculate olfactory bulbs are not known in fossil lungfish endocasts nor in tetrapodomorphs. The condition of the olfactory bulbs in fossil coelacanths is unknown, though the endocast of *Latimeria* belies the pedunculate nature of its olfactory bulbs.

## 4.5. Jaw musculature

For the taxa studied here across the fish–tetrapod transition, the region of greatest disparity between brain and endocast corresponds to a region of low muscle anatomical cross-sectional area. Specifically, in the tetrapods and lepidosirenid lungfish in particular, this occurs where there is space between the neurocranial cavity and the temporalis musculature (i.e. where the musculature is not attached to part of the frontoparietal or pterygoid bones). *Neoceratodus* appears to be an exception here as the region of greatest muscle cross-sectional area corresponds to a region of high disparity in the diencephalic region dorsally. Besides there being shrinkage artefacts in the adult specimen used herein, this pattern of brain–endocast disparity is consistent with the juvenile specimen studied by Clement *et al.* [15] and as such we consider the muscle–brain–endocast relationship seen at this point in *Neoceratodus* to be real. Rather than the brain being compromised in this region by such a mass of muscle, the brain cavity is actually reinforced and supported here by a dorsal expansion of the squamosal which contacts the supraorbital and, anterior to this, the dermosphenotic. In *Protopterus* spp. the squamosal is a long, slender element that does not contact the supraorbital and therefore offers little protection to the brain cavity.

As such, in *Neoceratodus*, there is probably less need for structural support and reinforcement in this region compared with the telencephalic region where the telencephalon is encased by ventrolateral projections of the frontoparietal and supported ventrally by the parasphenoid. In this region, the temporalis muscles are attached to both the median crista of the frontoparietal but also the pterygoid.

Although it is tempting to extrapolate correspondences in the location of mandibular adductors and endocast space in modern lungfishes and salamanders onto early tetrapods, we note that the organization of the jaw closure musculature in these two lineages is quite distinct from inferred organization of these muscles in early tetrapods. In early tetrapodomorphs, the adductor mass is laterally restricted to a narrow cavity between the palatoquadrate medially and the dermal cheek laterally, with the adductors originating on the palatoquadrate or cheek and inserting on the mandibular ramus [63]. In this organization, which persisted across the fin-to-limb transition, the mandibular adductors do not exert direct force on the neurocranium or vault, and the only force exerted on these elements would occur through normal reaction forces on tooth bearing bones of the anterior palate during the bite cycle or lateral forces during muscle compression [63]. In fact, expansion of the mandibular adductors to surround the braincase and vault entirely appears to be limited to modern batrachians [64], with earlier members of the putative lissamphibian total group retaining a small adductor compartment far lateral to the braincase [65,66]. This is in fact consistent with the generalized anatomy of the jaw adductor system in reptiles as well, although the expansion of the upper temporal fenestra in this group appears to correspond with a closer relationship between the vault and novel subdivisions of the mandibular adductor system [67].

How then our results might apply to patterns of coevolution between brain, braincase and mandibular adductor muscle mass, may be complex, likewise inference of early tetrapod brain size and shape from neurocranial endocasts. Our results show that, in sarcopterygians lacking an intracranial joint, endocast fidelity is greatest where neurocranial support is reinforced against lateral mandibular adductor muscle mass. The absence of a large mandibular adductor muscle mass lateral to the sphenethmoid portion of the braincase in basal tetrapods precludes the requirement for considerable ossified reinforcement in this region (figure 9). This is in line with qualitative

observations by Romer & Edinger [21] of the brain–endocast relationship in salamanders and frogs. Although we expect that early tetrapodomorphs with a patent intracranial joint may have exhibited *Latimeria*-like discrepancies between endocast and brain, closure of the intracranial joint at the fin-to-limb transition should permit a reliable inference of brain size and shape in early tetrapods.

# 5. Conclusion

We propose that the regions of the brain with the greatest distance from the enclosing neurocranium (represented by the endocast) arise from biomechanical constraints in sarcopterygian fish and salamanders. Here, we record a correspondence between the regions of greatest musculature in the head of the lungfish and the regions of closest fit of the brain to the neurocranium. This pattern holds for extant modern salamanders. Furthermore, the conservative nature of the basal tetrapod neurocranial bauplan, that is, an ossified otic region with the medial and anterior portion of an unossified neurocranium being supported by the parasphenoid, basisphenoid suggests that the endocranial cavity morphology may have been conserved from basal tetrapods to modern salamanders.

Ethics. Approval for the experimental procedures in this study were approved by the University of Edinburgh School of Geosciences Ethics Committee. They were conducted under schedule 1 (UK) at the University of Edinburgh, School of Biological Sciences.

Data accessibility. Micro-CT for *P. aethiopicus, P. dolloi, Ambystoma* and *Cynops* are publicly available on Figshare [68]. MRI data for *Neoceratodus forsteri* can be obtained from the Digital Fish Library catalogue, University of California San Diego (http://www.digitalfishlibrary.org). MRI data can be obtained on request from Dutel *et al.* [17].

Authors' contributions. T.J.C. and A.M.C. conceived the project, segmented the scan data, interpreted the data, produced the figures and wrote the manuscript. T.J.C. and J.D.P. stained specimens and generated scan data. T.J.C. analysed the data. All authors contributed to the writing of the manuscript and approved this final draft.

Competing interests. The authors declare no competing financial or non-financial interests.

Funding. Funding for T.J.C. was provided by Callidus Services Ltd, UK, and from a Flinders University International Research Fellowship. A.M.C. acknowledges funding from Australian Research Council grants DP 160102460 & DP 200103398, and thanks John Long (Flinders University) for supporting travel to the University of California San Diego.

Acknowledgements. We thank Larry Frank and Rachel Berquist (University of California, San Diego, http://csci.ucsd.edu) for the supply of the adult *Neoceratodus* scan from the Digital Fish Library (http://www.digitalfishlibrary.org), and thank Dianne Bray and Martin Gomon (Museum Victoria) for supplying comparative *Neoceratodus* material. We thank Hugo Dutel (University of Bristol) for supplying MRI scan data of adult *Latimeria*. Glenn Northcutt (University of California, San Diego) and Per Ahlberg (Uppsala University) are thanked for insightful discussions. Lionel Cavin and an anonymous reviewer are thanked for their constructive comments helping to improve this manuscript during review.

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
