## [Reviewer comments · Royal Society Open Science]

Review History

RSOS-200933.R0 (Original submission)

Review form: Reviewer 1 (Lionel Cavin)

Is the manuscript scientifically sound in its present form?

Yes

Are the interpretations and conclusions justified by the results?

Yes

Is the language acceptable?

Yes

Do you have any ethical concerns with this paper?

No

Have you any concerns about statistical analyses in this paper?

No

Recommendation?

Accept with minor revision (please list in comments)

Comments to the Author(s)

This is an interesting paper dealing with the relationships between the brain shape and the endocast in several sarcopterygian fishes and basal tetrapods. The authors show that there is a connection between the brain – endocast disparity and the ossification level of the braincase associated with to masticatory musculature in four of the six compared taxa, but not in *Latimeria* and *Neoceratodus*. Although they provide explanations why the two taxa display a different situation, the title of the paper is a bit too simplifier as it suggests that the connection between brain-endocast disparity and musculature is universal among sarcopterygian.

The explanation for the peculiar situation in *Latimeria* is straightforward because of its very special braincase architecture among Recent sarcopterygians with an intracranial joint. For *Neoceratodus*, the explanation for its organization different from the other taxa examined is a bit more uncertain. The authors state line 85 that the brain volume corresponds to 80% of the cranial cavity in a young individual, but 45% only in the adult examined in the present study (Table I). They attribute this shrinkage of the brain (in particular at the level of the telencephalon and olfactory tract region) as caused by preservation of the specimen in alcohol for a long. This might be true, but this explanation should not eliminate the possibility that the size might actually decrease in proportion during growth, as in *Latimeria* but to a lesser degree. The peculiar situation in *Neoceratodus* is also explained by the development of dermal ossifications that reinforced the brain cavity in this fish. It's a pity, however, that the *Neoceratodus* case study cannot be fully compared to the others taxa because of a methodological problem. This difficulty of interpretation is made visible lines 486 – 489, with the expression 'we consider...' used by the authors that should be replaced, in my sense, by 'we suggest...'. Anyway, this issue does not weaken the general interest of this research.

Another important issue is that brain size may vary in an important way intraspecifically between sexes of a single species, and even in a single individual over breeding cycle in teleosts, such as the sticklebacks (20% of volume may change) (Buechel et al., 2017), sunfish (axelrod et al., 2018), but also in frogs (Luo et al., 2017; Mai et al., 2017) and even in mammals with a shrinkage and re-growth of the skull and presumably of the brain during life cycle of the weasel (Dechmann et al., 2017). I suggest that this issue should be addressed in the discussion as it may have an implication on the hypothesis of a connection between brain shape and brain cavity, which would be affected mostly by the muscles arrangement.

Some specific comments:

- line 43. The split between *Neoceratodus* and the lepidosirenid lineage is said to be "more than 250 mya" based on Heinicke et al (2009). According to this paper, the split is supposedly pretty much older (277 mya), and an approximately similar age was also obtained on the basis of a palaeontological and morphological analysis by Kemp et al. (2017) (270 to 300 mya depending of the uncertainty of datation of some fossils).
- line 197. 'figure'
- line 292 'a different'
- line 340-341: This is not so obvious why the situation should be considered as analogous rather than homologous. Is it only because of the difference between tract and peduncles? Do we have clues visible in fossils of piscine sarcopterygians situated "in between" *Latimeria* and *Neoceratodus* that may shed light on the evolutionary trajectory of this feature?
- line 381. delete the reference after [41]
- line 476: please add a reference at the end of this §.

References

- Axelrod, C. J., Laberge, F., & Robinson, B. W. (2018). Intraspecific brain size variation between coexisting sunfish ecotypes. *Proceedings of the Royal Society B*, 285(1890), 20181971.
- Buechel, S. D., Noreikiene, K., DeFaveri, J., Toli, E., Kolm, N., & Merilä, J. (2019). Variation in sexual brain size dimorphism over the breeding cycle in the three-spined stickleback. *Journal of Experimental Biology*, 222(7).

Dechmann, D. K., LaPoint, S., Dullin, C., Hertel, M., Taylor, J. R., Zub, K., & Wikelski, M. (2017). Profound seasonal shrinking and regrowth of the ossified braincase in phylogenetically distant mammals with similar life histories. *Scientific reports*, 7, 42443.

Kemp, A., Cavin, L., & Guinot, G. (2017). Evolutionary history of lungfishes with a new phylogeny of post-Devonian genera. *Palaeogeography, Palaeoclimatology, Palaeoecology*, 471, 209-219.

Luo, Y., Zhong, M. J., Huang, Y., Li, F., Liao, W. B., & Kotschal, A. (2017). Seasonality and brain size are negatively associated in frogs: evidence for the expensive brain framework. *Scientific reports*, 7(1), 1-9.

Mai, C. L., Liao, J., Zhao, L., Liu, S. M., & Liao, W. B. (2017). Brain size evolution in the frog *Fejervarya limnocharis* supports neither the cognitive buffer nor the expensive brain hypothesis. *Journal of Zoology*, 302(1), 63-72.

Lionel Cavin

Review form: Reviewer 2

Is the manuscript scientifically sound in its present form?

Yes

Are the interpretations and conclusions justified by the results?

Yes

Is the language acceptable?

Yes

Do you have any ethical concerns with this paper?

No

Have you any concerns about statistical analyses in this paper?

No

Recommendation?

Accept with minor revision (please list in comments)

Comments to the Author(s)

General remarks:

The changes of brain morphology from water to land is little known however is important to our understanding of sarcopterygian evolution and tetrapod origin since Devonian period. Challands et al. provides a new angle for investigating the changes that occurred over the fish-tetrapod transitions. Their work investigates six basal sarcopterygians as a proxy for quantifying the brain-braincase relationship, and highlights the importance of detailed understanding of the relationship between brain and endocast, which will avoid erroneous interpretations of physiology and/or behaviour based on the endosseous representations of functional units alone. The article is logical, well-written, and can inspire the future research on the brain evolution of sarcopterygians.

Minor comments:

Line 207, "(3) the endocasts of the lepidosirend lungfishes, *Ambystoma* and *Cynops* do not exhibit a distinct hypophyseal fossa". Do authors know the condition of the hypophyseal fossa in other basal tetrapods? According to the endocast of fossil amphibian *Edops* (Romer 1942), the hypophyseal is distinct.

Lines 277&278, delete (2015)

Line 359, "The basicranial musculature is not seen in lungfish or tetrapods", do authors want to explain why the "basicranial musculature is not seen in lungfish or tetrapods"? (due to the loss of intracranial joint in lungfish and tetrapods?)

Line 364, What do "early sarcopterygians" mean?

Line 368 "non-digited tetrapodomorphs." finned tetrapodomorphs is better.

Line 381 Delete (Neenan et al. 2014)

Figure 1 and others. specimen name should be Italic

Figure 9 The label for hyf is not clear shown

Other comments:

"hypophyseal fossa" or "hypophyseal fossae", should be consistent

Salamanders specimens for investigation were adults or juvenile?

Decision letter (RSOS-200933.R0)

Dear Dr Challands

On behalf of the Editors, I am pleased to inform you that your Manuscript RSOS-200933 entitled "Mandibular musculature constrains brain-endocast disparity in sarcopterygians" has been accepted for publication in Royal Society Open Science subject to minor revision in accordance with the referee suggestions. Please find the referees' comments at the end of this email.

The reviewers and handling editors have recommended publication, but also suggest some minor revisions to your manuscript. Therefore, I invite you to respond to the comments and revise your manuscript.

- Ethics statement

- Data accessibility

<http://datadryad.org/submit?journalID=RSOS&manu=RSOS-200933>

- Competing interests

- Authors' contributions

- Acknowledgements

- Funding statement

Because the schedule for publication is very tight, it is a condition of publication that you submit the revised version of your manuscript before 14-Aug-2020. Please note that the revision deadline will expire at 00.00am on this date. If you do not think you will be able to meet this date please let me know immediately.

- 1) A text file of the manuscript (tex, txt, rtf, docx or doc), references, tables (including captions) and figure captions. Do not upload a PDF as your "Main Document";

- 2) A separate electronic file of each figure (EPS or print-quality PDF preferred (either format should be produced directly from original creation package), or original software format);
- 3) Included a 100 word media summary of your paper when requested at submission. Please ensure you have entered correct contact details (email, institution and telephone) in your user account;
- 4) Included the raw data to support the claims made in your paper. You can either include your data as electronic supplementary material or upload to a repository and include the relevant doi within your manuscript. Make sure it is clear in your data accessibility statement how the data can be accessed;
- 5) All supplementary materials accompanying an accepted article will be treated as in their final form. Note that the Royal Society will neither edit nor typeset supplementary material and it will be hosted as provided. Please ensure that the supplementary material includes the paper details where possible (authors, article title, journal name).

If your manuscript is newly submitted and subsequently accepted for publication, you will be asked to pay the article processing charge, unless you request a waiver and this is approved by Royal Society Publishing. You can find out more about the charges at <https://royalsocietypublishing.org/rsos/charges>. Should you have any queries, please contact openscience@royalsociety.org.

on behalf of Professor Marcelo Sanchez (Associate Editor) and Kevin Padian (Subject Editor)
openscience@royalsociety.org

Reviewer comments to Author:
Reviewer: 1

Comments to the Author(s)

This is an interesting paper dealing with the relationships between the brain shape and the endocast in several sarcopterygian fishes and basal tetrapods. The authors show that there is a connection between the brain – endocast disparity and the ossification level of the braincase associated with to masticatory musculature in four of the six compared taxa, but not in *Latimeria* and *Neoceratodus*. Although they provide explanations why the two taxa display a different

situation, the title of the paper is a bit too simplifier as it suggests that the connection between brain-endocast disparity and musculature is universal among sarcopterygian.

The explanation for the peculiar situation in *Latimeria* is straightforward because of its very special braincase architecture among Recent sarcopterygians with an intracranial joint. For *Neoceratodus*, the explanation for its organization different from the other taxa examined is a bit more uncertain. The authors state line 85 that the brain volume corresponds to 80% of the cranial cavity in a young individual, but 45% only in the adult examined in the present study (Table I). They attribute this shrinkage of the brain (in particular at the level of the telencephalon and olfactory tract region) as caused by preservation of the specimen in alcohol for a long. This might be true, but this explanation should not eliminate the possibility that the size might actually decrease in proportion during growth, as in *Latimeria* but to a lesser degree. The peculiar situation in *Neoceratodus* is also explained by the development of dermal ossifications that reinforced the brain cavity in this fish. It's a pity, however, that the *Neoceratodus* case study cannot be fully compared to the others taxa because of a methodological problem. This difficulty of interpretation is made visible lines 486 – 489, with the expression 'we consider...' used by the authors that should be replaced, in my sense, by 'we suggest...'. Anyway, this issue does not weaken the general interest of this research.

Another important issue is that brain size may vary in an important way intraspecifically between sexes of a single species, and even in a single individual over breeding cycle in teleosts, such as the sticklebacks (20% of volume may change) (Buechel et al., 2017), sunfish (axelrod et al., 2018), but also in frogs (Luo et al., 2017; Mai et al., 2017) and even in mammals with a shrinkage and re-growth of the skull and presumably of the brain during life cycle of the weasel (Dechmann et al., 2017). I suggest that this issue should be addressed in the discussion as it may have an implication on the hypothesis of a connection between brain shape and brain cavity, which would be affected mostly by the muscles arrangement.

Some specific comments:

- line 43. The split between *Neoceratodus* and the lepidosirenid lineage is said to be "more than 250 mya" based on Heinicke et al (2009). According to this paper, the split is supposedly pretty much older (277 mya), and an approximately similar age was also obtained on the basis of a palaeontological and morphological analysis by Kemp et al. (2017) (270 to 300 mya depending of the uncertainty of datation of some fossils).
- line 197. 'figure'
- line 292 'a different'
- line 340-341: This is not so obvious why the situation should be considered as analogous rather than homologous. Is it only because of the difference between tract and peduncles? Do we have clues visible in fossils of piscine sarcopterygians situated "in between" *Latimeria* and *Neoceratodus* that may shed light on the evolutionary trajectory of this feature?
- line 381. delete the reference after [41]
- line 476: please add a reference at the end of this §.

References

- Axelrod, C. J., Laberge, F., & Robinson, B. W. (2018). Intraspecific brain size variation between coexisting sunfish ecotypes. *Proceedings of the Royal Society B*, 285(1890), 20181971.
- Buechel, S. D., Noreikiene, K., DeFaveri, J., Toli, E., Kolm, N., & Merilä, J. (2019). Variation in sexual brain size dimorphism over the breeding cycle in the three-spined stickleback. *Journal of Experimental Biology*, 222(7).
- Dechmann, D. K., LaPoint, S., Dullin, C., Hertel, M., Taylor, J. R., Zub, K., & Wikelski, M. (2017). Profound seasonal shrinking and regrowth of the ossified braincase in phylogenetically distant mammals with similar life histories. *Scientific reports*, 7, 42443.
- Kemp, A., Cavin, L., & Guinot, G. (2017). Evolutionary history of lungfishes with a new phylogeny of post-Devonian genera. *Palaeogeography, Palaeoclimatology, Palaeoecology*, 471, 209-219.

Luo, Y., Zhong, M. J., Huang, Y., Li, F., Liao, W. B., & Kotschal, A. (2017). Seasonality and brain size are negatively associated in frogs: evidence for the expensive brain framework. *Scientific reports*, 7(1), 1-9.

Mai, C. L., Liao, J., Zhao, L., Liu, S. M., & Liao, W. B. (2017). Brain size evolution in the frog *Fejervarya limnocharis* supports neither the cognitive buffer nor the expensive brain hypothesis. *Journal of Zoology*, 302(1), 63-72.

Lionel Cavin

Reviewer: 2

Comments to the Author(s)

General remarks:

The changes of brain morphology from water to land is little known however is important to our understanding of sarcopterygian evolution and tetrapod origin since Devonian period. Challands et al. provides a new angle for investigating the changes that occurred over the fish-tetrapod transitions. Their work investigates six basal sarcopterygians as a proxy for quantifying the brain-braincase relationship, and highlights the importance of detailed understanding of the relationship between brain and endocast, which will avoid erroneous interpretations of physiology and/or behaviour based on the endosseous representations of functional units alone. The article is logical, well-written, and can inspire the future research on the brain evolution of sarcopterygians.

Minor comments:

Line 207, "(3) the endocasts of the lepidosirend lungfishes, *Ambystoma* and *Cynops* do not exhibit a distinct hypophyseal fossa". Do authors know the condition of the hypophyseal fossa in other basal tetrapods? According to the endocast of fossil amphibian *Edops* (Romer 1942), the hypophyseal is distinct.

Lines 277&278, delete (2015)

Line 359, "The basicranial musculature is not seen in lungfish or tetrapods", do authors want to explain why the "basicranial musculature is not seen in lungfish or tetrapods"? (due to the loss of intracranial joint in lungfish and tetrapods?)

Line 364, What do "early sarcopterygians" mean?

Line 368 "non-digited tetrapodomorphs." finned tetrapodomorphs is better.

Line 381 Delete (Neenan et al. 2014)

Figure 1 and others. specimen name should be Italic

Figure 9 The label for *hyf* is not clear shown

Other comments:

"hypophyseal fossa" or "hypophyseal fossae", should be consistent

Salamanders specimens for investigation were adults or juvenile?

Author's Response to Decision Letter for (RSOS-200933.R0)

See Appendix A.

Decision letter (RSOS-200933.R1)

Dear Dr Challands,

It is a pleasure to accept your manuscript entitled "Mandibular musculature constrains brain-endocast disparity between sarcopterygians" in its current form for publication in Royal Society Open Science.

on behalf of Professor Marcelo Sanchez (Associate Editor) and Kevin Padian (Subject Editor)
openscience@royalsociety.org

Appendix A

Challands, Pardo & Clement Response to referees.

We thank both reviewers for their time and comments in improving our manuscript

Reviewer 1

Reviewer 1 comments “..the title of the paper is a bit too simplifier as it suggests that the connection between brain-endocast disparity and musculature is universal among sarcopterygian.”. In response to this we have changed the title to *Mandibular musculature constrains brain-endocast disparity between sarcopterygians*.

We have added a sentence at line 487 indicating that brain-endocast disparity between juvenile and adult *Neoceratodus* could be due to a similar process as seen in *Latimeria*. We also have changed the word ‘consider’ to ‘suggest’ as recommended by Reviewer 1 but also state that given the lack of intracranial joint in *Neoceratodus* and that the shrinkage in the brain of the adult specimen is similar to that seen in the experimental data of Carlisle et al. (2017) we indicate that alcohol shrinkage is our preferred explanation for this observation.

We have added to the discussion a section on interspecific and intersex variation in brain size in teleosts and tetrapods using the references kindly provided by Reviewer 1.

Line 43: ‘..more than 250 mya..’ changed to ‘..277 mya..’.

Line 292: Typo corrected.

Line 340-341:

Line 381: Reference deleted.

Line 476: Lu et al. (2016) added.

Reviewer 2

Reviewer 2 - Line 207, “3) the endocasts of the lepidosirend lungfishes, *Ambystoma* and *Cynops* do not exhibit a distinct hypophyseal fossa”. Do authors know the condition of the hypophyseal fossa in other basal tetrapods? According to the endocast of fossil amphibian *Edops* (Romer 1942), the hypophyseal is distinct.

Response - We have inserted the following sentence at line 472 with additional references that address Reviewer 2’s comment on the hypophyseal fossa in early tetrapods. “The hypophyseal fossa itself has been described in some detail across early tetrapods [24, 38], and is relatively deep in early tetrapods outside of the tetrapod crown [21, 24, 25, 38, 39, 53].”

Reviewer 2 - Line 359, “The basicranial musculature is not seen in lungfish or tetrapods”, do authors want to explain why the “basicranial musculature is not seen in lungfish or tetrapods”? (due to the loss of intracranial joint in lungfish and tetrapods?)

Response - We have inserted the following at this point (now line 360) to clarify this query: “...as a consequence of the loss of the intracranial joint...”

Reviewer 2 - Line 364, What do “early sarcopterygians” mean?

Response – We have changed ‘early’ to ‘basal’ to avoid ambiguity in meaning here.

Reviewer 2 - Line 368 “non-digitated tetrapodomorphs.” finned tetrapodomorphs is better.

Response – Changed.

Reviewer 2 - Line 381 Delete (Neenan et al. 2014)

Response – Changed.

Reviewer 2 - Figure 1 and others. specimen name should be Italic

Response – Taxa names have been removed from figures as they are included in the figure captions.

Reviewer 2 - Figure 9 The label for hyf is not clear shown.

Response – We have now labelled the hypophyseal fossa in Figure 9 a.

Reviewer 2 - “hypophyseal fossa” or “hypophyseal fossae”, should be consistent.

Response – Hypophyseal fossae is the plural of hypophyseal fossae and is used in the correct manner in our manuscript. Whereas our inclusion of ‘hypophyseal fossae’ is grammatically correct on line 470 we have changed the sentence to make it read better.

Reviewer 2 - hypophyseal fossa” or “hypophyseal fossae”, should be consistent

Response – Fossa used now.

Reviewer 2 - Salamanders specimens for investigation were adults or juvenile?

Response – ‘...adult individuals..’ added on line 137 to clarify.

Other changes made

Author affiliation: changed “School of Biological Sciences” from Clement affiliation to “College of Science and Engineering” (university restructuring).

Line 176: Removed opening parenthesis.

Line 197: Corrected spelling of ‘figure’.

Line 278: removed ‘(2015)’.

Line 357: Corrected spelling of ‘configuration’.

Line 453: ‘the’ added to make grammatical sense.

Line 472: Space added between sentences where previously missing.

Line 476: ‘it’ added to make grammatical sense.

Taxon names and the text on RHS on Figs 1 & 2

Formatting of hyperlinks corrected.

Formatting of references corrected to journal style.

Double carriage returns eliminated throughout where present.

Acknowledgments made to the reviewers: *“Lionel Cavin and an anonymous reviewer are thanked for their constructive comments helping to improve this manuscript during review.”*